# Interethnic Influencing Factors Regarding Buttocks Body Image in Women from Nigeria, Germany, USA and Japan

**DOI:** 10.3390/ijerph192013212

**Published:** 2022-10-14

**Authors:** Christoph Wallner, Svenja Kruber, Sulaiman Olanrewaju Adebayo, Olusola Ayandele, Hikari Namatame, Tosin Tunrayo Olonisakin, Peter O. Olapegba, Yoko Sawamiya, Tomohiro Suzuki, Yuko Yamamiya, Maximilian Johannes Wagner, Marius Drysch, Marcus Lehnhardt, Björn Behr

**Affiliations:** 1Department of Plastic Surgery, BG University Hospital Bergmannsheil, Ruhr University Bochum, Bürkle-de-la-Camp Platz 1, 44789 Bochum, Germany; 2Department of Psychology and Behavioural Studies, Ekiti State University, Ado Ekiti 360102, Nigeria; 3Department of Psychology, University of Ibadan, Ibadan 200005, Nigeria; 4Department of General Studies, The Polytechnic, Ibadan 200285, Nigeria; 5Faculty of Human Sciences, University of Tsukuba, Ibaraki 305-8577, Japan; 6Faculty of Humanities, North West University, Mafikeng 2790, South Africa; 7Department of Child Psychology, Tokyo Future University, Tokyo 120-0023, Japan; 8Department of Undergraduate Studies, Temple University, Japan Campus, Tokyo 154-0004, Japan

**Keywords:** social media, female, buttocks, body image, WHR

## Abstract

Background: Body image research deals a lot with awareness of the body as an entity. Studies that consider individual anatomical aspects and place them in an intercultural context are rarely present. Methods: For this purpose, general data, body perception and judgment of body images from 2163 (48% female and 52% male) participants from Germany, Nigeria, the USA and Japan were evaluated as part of a survey. Results: There were clear differences in the personal body image of the participants’ own buttocks, the buttocks as a beauty ideal and the way in which dissatisfaction was dealt with in different countries. In addition to sexual well-being (importance score: 0.405 a.u.), the country of origin (0.353), media consumption (0.042) and one’s own weight (0.069) were also identified as influencing factors for satisfaction with one’s own buttocks. A clear evolution could be derived regarding a WHR (waist-to-hip ratio) of well below 0.7, which was consistently favored by the participants but also propagated by influencers through images (*p* < 0.001). In this context, participants who indicated celebrities as role models for the buttocks showed a correspondingly high level of dissatisfaction with their own buttocks (R = −0.207, *p* < 0.001, ρ = −0.218). Conclusion: Overall, a highly significant correlation was shown between the consumption frequency of Instagram, TikTok and pornography with the negative perception of women’s own buttocks.

## 1. Introduction

Sociocultural perspectives on body image assert that culture influences attitudes, behaviors and values regarding the body [1]. A woman’s attitude toward her own body is influenced by what she perceives to be physically attractive in Western cultures [2]. Media can be a vehicle for the transportation of an idealized body. One way media exposure influences body image is through appearance-based comparisons with popular media. This leads to body dissatisfaction because many women focus their attention on how their own appearances do not equate with idealized media [3]. Looking at images that correspond to the current ideal of beauty leads to negative effects on women’s own body image [1]. There is a body of evidence showing that media consumption engenders a higher rate of body dissatisfaction and ultimately eating disorders in women. A range of negative psychological and physical health outcomes are associated with dissatisfaction, such as social anxiety, interest in esthetic surgery and lower life satisfaction [4,5,6].

In White-centered mainstream media, thin-ideal images of women’s bodies are ubiquitous and widely advertised across various media channels, including magazines, social media and television [7,8,9]. Studies showed that beauty trends conveyed via social media have an enormous influence on a woman’s self-image, not only in Western but also in Asian cultures [10,11]. In particular, image-based social media channels, such as Instagram, present a monotonous body image that produces significant pressure on social media users. It was shown that exposure to attractive celebrities or peers increases body dissatisfaction and negative mood. This impact was interceded by a state appearance comparison [12].

With the thinspiration trend on social media, a certain body picture was enforced [8]. Most pictures of females within this trend are muscular and toned. Fitspiration content places a greater emphasis on muscular physique than thinspiration or bonespiration, which places more emphasis on the thin ideal [8].

Recently, various trends have brought the curvy figure to the center of attention. This trend can be seen through increasing frequencies of the hashtags #thick, #thicc and #slimthick [13]. The term curvy is not clearly defined and is understood in the media as a combination of a slim waist and wider hips but can also refer to plus-sized and overweight figures [14,15]. It can also be understood as an exaggerated hourglass figure with a slim waist, large breasts and a wide pelvis [16]. A quantitative statement about this hourglass figure was made in a study by Hunter et al. in terms of a waist-to-hip ratio (WHR) below 0.7 [17]. Although research on women’s body image has not been conducted on slim-thick or hourglass bodies, this type of body may be as unrealistic and harmful to women’s body image as a thin-ideal type. Instead of calling this type of body plus-sized, average or curvy-slim, which all carry different meanings, McCombs suggested naming it “slim-thick-ideal”, combining an unrealistic exaggerated image of a flat and thin waist with a large circumference around the hips and breasts [13].

A foundation of female attractiveness is the WHR in conjunction with the BMI (body mass index). Although the BMI and the WHR are seen as competing indicators of female attractiveness, the slim-thick trend in particular counters this argument [18,19,20,21]. From evolutionary psychology, it can be assumed that the WHR has not only developed as a factor of gender discrimination but also to distinguish pregnant from not pregnant and, therefore, represents an important attractiveness feature. In the current scientific discussion, the WHR is understood as a backup signal, i.e., a signal that is accompanied by other signals with a similar message and, therefore, underlines attractiveness [22,23]. Therefore, an assumption can be made that the WHR is a universal transcultural dogma. Actually, a preference for a WHR of 0.7 was shown in many cultures [21,24,25,26]. A study of pre-industrial people in Tanzania showed a preference for heavier women and the WHR played a minor role. It was concluded that when there is a surplus of food resources, the WHR as an attractiveness trait comes into play [27]. Further studies showed that, ultimately, a WHR of around 0.7 is also preferred in pre-industrial groups, provided that this is not only reflected in a frontal image but also from the side. Here, steatopygia, i.e., obesity in the coccyx, is perceived as very attractive and also has an influence on a low WHR [28,29]. In summary, the female WHR can be used as an attractiveness trait and a value of approximately 0.7 can be confirmed as attractive in many cultural backgrounds.

Unlike before, this slim-thick-ideal type was not transported through White-centric media first but appeared first among other ethnic communities, such as Latino and Black communities [13]. A curvy female is more desired than a slim woman in Latin culture [30]. A similar finding was also discovered among African American women, who find the thin-ideal type less desirable and do not internalize them as much as White women [31]. Furthermore, young African American women not only find a curvy woman’s body more beautiful than a thin one but also describe the latter as “being for white women” [32]. Based on this, a preference for a curvy body by different ethnic groups can be observed and is presented as a Westernized form in the slim-thick-ideal type [13].

The unrealistic shape is reflected with great interest on social media. Celebrities such as Cardi B or Kim Kardashian achieve these forms only through esthetic procedures, such as breast augmentation, liposuction of the waist and especially buttock augmentation. The waist and buttocks represent the fundamental components of the WHR. In particular, the buttocks are pushed into the center of attention. This temporal trend for certain esthetic interventions can be clearly linked to statements by influencers. For example, there were significantly more searches on Google for BRCA mutations and breast reconstruction following Angelina Jolie’s statement and Kim Kardashian’s buttock augmentation [33]. In addition to this influence of celebrities, the risk of these interventions must not be downplayed. With a mortality rate of 1:3000 to 1:6000, buttock augmentation by fat transfer is one of the procedures in esthetic surgery with the highest mortality [34]. Unlike the ideal-fit or ideal-thin types, the slim-thick-ideal type is very difficult to achieve naturally. Studies show that exposure to this type of body image leads to higher surveillance, reflection on one’s own body image and lower body self-esteem [15,35]. However, this interaction with slim-thick-ideal imagery is still a topic of ongoing discussion [36]. Nevertheless, it is argued that this body image is potentially more harmful than others and will be exacerbated by exposure and comparison to slim-thick imagery [13,15].

So far, only the ethnic difference in the evaluation of body image evaluation has been described. However, there are also differences in self-evaluation regarding certain character traits, such as perfectionism, which leads to increased self-optimization, comparison with the peer group and body dissatisfaction [37]. Two different approaches are described for body perfectionism: worrying about imperfection and hope for perfection [13]. People with higher physical appearance perfectionism are more vulnerable to social media comparison [38].

### 1.1. The Present Study

How much social media impacts one’s own body perception is still a topic of current scientific discussion [39]. In the mixed situation between inflationary social media influence, increasing body image uncertainty and increasing self-optimization with esthetic surgery, individual persuading forces have not been considered so far. The buttocks as the focal point of this slim-curvy ideology have rarely been studied in correspondence with the WHR. Furthermore, the influence of specific influencers or role models on the assessment of one’s own buttocks has not yet been examined. To overcome the difficulty of being biased by a country’s peer group, the intention was to broaden the base by surveying groups from several countries with different ethnicities and diverse access to Western culture.

The current study was designed to broaden the literature about the perception of female buttocks with the implication of social media influence. The extent to which media influence the perception of a very specific part of the body should be examined. In addition, questions arise regarding at what level media exposure influences satisfaction with an anatomical feature.

Although culturally isolated from the rest of the world for a long time, Japan exhibits a certain Westernization of body image. Nevertheless, in Japan, the ideal-thin trend is still much more emphasized than the ideal-fit trend [40]. Ando described juxtaposed emancipating body images in the female Japanese: the ideal-thin type is transported within the “hattou-shin” beauty concept and “kawaii” illustrates the naïve objectified female. A third form of esthetics represents the young woman between youth and adulthood and is called “shôjo” [41,42]. As in Japanese culture, beauty standards are also influenced by spiritual, religious and cultural frameworks, which conflict with the Western ideal; the implication of this cultural peer group has an important role in this study. As there is a growing influence of Western beauty standards, there is still a Japanese-specific reaction to these influences [42]. There is scientific evidence that body images dictated by the media influence Japanese women and put them under pressure [40,43]. On the other hand, as one of the most populous countries in Africa, Nigeria is inhomogeneous regarding Western influence due to its varying access to media in the country. However, an increasingly widespread Westernization of the ideal of beauty in recent times in Nigeria was documented [44,45]. This heterogeneity of cultural backgrounds between the USA, Germany, Nigeria and Japan can provide information about the different approaches to dealing with the anatomical attributes of the buttocks.

### 1.2. Hypotheses

**Hypothesis** **1a** **(H1a).***We hypothesized a different view of the body with an emphasis on the buttocks between countries. With this different approach, a different way of dealing with dissatisfaction in the form of the urge for optimization is likewise assumed*.

**Hypothesis** **1b** **(H1b).***In addition to the large cultural differences, we wanted to identify other psycho-social factors influencing satisfaction with one’s own buttocks*.

**Hypothesis** **2a** **(H2a).***We hypothesized the interaction with imagery media as one major driving force for the satisfaction with one’s own buttocks*.

**Hypothesis** **2b** **(H2b).***In addition, we assumed a stringent specification of beauty ideals of the buttocks through social media. This could be verified by matching fitness influencers to study participants’ preferred WHRs*.

## 2. Materials and Methods

### 2.1. Study Design

A cross-sectional study was conducted by facilitating a web-based questionnaire with several sub-categories that depict the most discussed elements in gluteal esthetics. Recruitment took place between 1 January 2022 and 30 June 2022. The first part contained general questions about demography, level of education, relationship status and social media consumption, as well as feelings about one’s own buttocks, one’s own body preference and one’s attitude toward esthetic interventions. The second part contained sets of different series of images, each with a morphed property in a lifelike 3D model. The questionnaire was entered into the SurveyMonkey (San Mateo, CA, USA) system. The survey was translated into German, English and Japanese, and participants were able to select the language. The survey is added in the Appendix A to show the included variables in detail.

The survey was sent to participants in Nigeria, Germany, USA and Japan by local collaborators. The study was approved by the Institutional Review Board of the University of Tsukuba (protocol code 2022-28A, date of approval: 2022-05-18).

The 3D models were generated and morphed in DAZ Studio 3D (version: 4.12.1.117; DAZ 3D, Inc., Salt Lake City, UT, USA). The morphed pictures were randomly shuffled for each respondent to avoid bias. The following parameters were separately morphed into the image panels: proportions of frontal WHR (0.6, 0.7, 0.8) and lateral WHR (0.6, 0.7, 0.8). This method was recently published [46].

### 2.2. IP Classification and Verification of Origin

Within the SurveyMonkey service, the IP address of each participant was collected anonymously via IP tracking. To verify the origin country of every participant and rule out bots and scam IP addresses, the IP addresses given by SurveyMonkey were linked with the original location using the Python library urlib.request by utilizing the website https://geolocation-db.com.

### 2.3. Classification and Regression Tree (CART) Analysis

Classification and regression tree (CART) analysis was used for binary recursive partitioning in Python (version: 3.8.3). Pandas (version: 1.3.0) and NumPy (version: 1.22.0) were implemented for the data handling. To evaluate the feature importance correlation, attribute evaluation was utilized. For the decision tree classifier (version: 1.0.1, scikit-learn), a maximal depth of 3 was chosen with a training size of 0.9 and a testing size of 0.1.

### 2.4. Instagram Influencer Data Collection

Instagram influencer stats were collected during April 2022 with the website https://starngage.com/. The aim was to collect stats of the top-followed 50 Instagram fitness influencers from the target countries Germany, Nigeria, USA and Japan. The mean WHR was calculated with at least three posted pictures in a front view.

### 2.5. Statistics and Data Presentation

For the statistical tests (ANOVA, Student’s *t*-test, Spearman test, Tukey’s and Dunnett’s T3 post hoc test, Pearson correlation), IBM SPSS Statistics 21 (version: 21.0.0.0; IBM, Armonk, NY, USA) was utilized. Graphs, charts and infographics were generated in Adobe Illustrator (version: 26.3.1; Adobe, Inc., San José, CA, USA) and Adobe Photoshop (23.4.1; Adobe, Inc., San José, CA, USA). The alpha level (α) was set at 0.05 when applicable.

## 3. Results

### 3.1. Survey Population

Participants from Germany (*n* = 321), Nigeria (*n* = 1172), USA (*n* = 366) and Japan (*n* = 304) were included and tracked to ensure that the correct IP address corresponded with the stated origin. The cohorts were surveyed online. Of the 2163 total participants, 111 did not complete the survey (completion rate: 94.9%). The median time to complete the survey was 4.47 min.

The study included 1042 females (48.2%), 1107 males (51.2%), 4 transsexuals (<1%) and 10 participants with another gender (<1%). The female proportion was 56.7% in Germany, 45.2% in Nigeria, 50% in the USA and 48.4% in Japan. The male proportion was 41.4% in Germany, 54.4% in Nigeria, 50% in the USA and 50.3% in Japan. The remaining genders were below 0.5%. The mean ages were 1.67 (equivalent to 18–24 years) in Nigeria, 2.06 (equivalent to 25–34 years) in Germany, 2.74 (equivalent to 25–34 years) in the USA and 3.43 (equivalent to 34–44 years) in Japan.

Due to the strong cooperation, we were able to gain the largest study population by far in Nigeria, with the youngest population in the study. More female participants were found in Germany and more males were found in Nigeria. The demographics are illustrated in Figure 1.

### 3.2. Perception of One’s Own Buttocks in Women

As a first step, the general results regarding each participant’s perception of their buttocks were evaluated. Participants were asked whether they were happy with their buttocks, would surgically change their buttocks due to esthetic reasons or had surgery done. Concerning happiness with one’s own buttocks, women from Germany gave a rating of 3.45, Nigeria 4.37, USA 3.66 and Japan 2.23 on average on a scale from 0 to 5. This was shown to be highly significantly different (*p* < 0.001). The proportions of women who answered the question regarding whether they would surgically change their buttocks because of esthetic reasons with yes were 23.8% in Germany, 8.2% in Nigeria, 63.1% in the USA and 4.3% in Japan (*p* < 0.001). Regarding the question regarding whether the women had esthetic procedures done on their buttocks, 17.2% of German women, 5.3% of Nigerian women, 11.2% of US women and 1.3% of Japanese women answered yes (*p* = 0.001). The women were also asked what part of a woman’s body they are most attracted to. Breasts were named by 21% in Germany, 20% in Nigeria, 29% in the USA and 28% in Japan. Buttocks were named by 20% in Germany, 34% in Nigeria, 57% in the USA and 5% in Japan. The abdomen was named by 11% in Germany, 5% in Nigeria, 0% in the USA and 3% in Japan as the first choice. Legs were named by 9% in Germany, 2% in Nigeria, 0% in the USA and 23% in Japan. The face was named by 39% in Germany, 39% in Nigeria, 15% in the USA and 41% in Japan as the most attractive part of the female body (*p* < 0.001). This is shown in Table 1.

As a first observation, Japanese women were significantly unhappier with their own buttocks compared with the other countries. The happiest were found in Nigeria. Despite this, Japanese women showed the least desire for buttock surgery by far with 4.3%. In contrast, almost two-thirds of the US female participants considered esthetic buttock procedures. German women demonstrated the highest rate of buttock surgery performed with 17.2%. Taken together, the countries with the lowest happiness did not necessarily lead to the highest desire for surgery or performed surgery. One explanation could be the low appreciation of the buttocks as the most attractive part of the female body in Japan at 5% compared with the USA at 57% or Germany at 20%. Broken down by country, the trend did not show that unhappier participants would be more likely to consider surgery. However, looking at the overall distribution of happiness according to conversion to consideration or surgery, there was a clear trend that unhappier individuals were more likely to consider or have had surgery. The conversion flows are shown in Figure 2.

Hypothesis 1a: The study presented a different view on the buttocks and coping with dissatisfaction in different countries.

### 3.3. Identification of Major Factors Influencing the Perception of One’s Own Buttocks

Since it was shown above that unhappiness with the buttocks was associated with a significantly increased desire for an operative change, the reasons are examined more closely here. For this reason, a CART analysis was carried out, which, based on happiness, sorts all the parameters asked according to their influencing factor. To achieve better meaningfulness here, the dimensionality of the variable happiness was reduced from a 5-star scale to yes and no (1, 0). The splitting point was the neutral value (3). Figure 3 shows the CART visualization, together with the entropy and fraction at every leaf node. 

Satisfaction with one’s own sexual life was shown to be the most influential factor, closely followed by country of origin, weight, level of education, ethnicity and frequency of TikTok consumption. Low satisfaction with one’s own sexual life below 1.5 stars on a 5-star panel was most significantly related to low happiness with one’s own buttocks. As shown before, Japan as a country of origin was linked with a lower happiness rating. In this subgroup, women with Black/African, Hispanic or Caucasian ethnicity had a better rating compared with women with Asian or Native American ethnicity. A lower education level and more frequent consumption (more than once a month) were correlated with a lower rating of happiness with one’s own buttocks. The importance of the factors is shown in Table 2.

Hypothesis 1b: This algorithm enabled identifying major forces that led to lower satisfaction with one’s own buttocks.

### 3.4. Social Media Was Identified as a Crucial Trigger for the Perception of Gluteal Esthetics in Females

As was shown above, TikTok seemed to have a major role in influencing satisfaction with one’s own buttocks in females across countries, age groups and ethnicities. Therefore, the participants were asked what their main role models for the perception of the ideal buttocks are. Multiple answers were possible, and the following choices were given: none, social media influencers, celebrities, fitness trainers and pornography. Cohen suggests that effect sizes can be interpreted as small (0.20), moderate (0.50) or large (0.80). These values corresponded to Pearson correlation factor (R) values of 0.10, 0.24 and 0.37, respectively [1,47].

Of the 963 total female participants, 212 (22%) expressed no specific influence on their personal gluteal perception, while 422 (44%) identified social media, 260 (27%) celebrities, 344 (36%) fitness role models and 29 (<1%) models from pornography as their main influence for one’s own gluteal perception. Women who stated no specific influencing factor demonstrated a positive and significant correlation R (Pearson correlation factor) = 0.102, *p* = 0.002 and ρ (Spearman rank correlation) = 0.106 with higher happiness with their own buttocks. A negative correlation between celebrities as stated role models and happiness was statistically significant (R = −0.207, *p* < 0.001, ρ = −0.218). On the other hand, the subjective influencing factors of social media, fitness role models or pornography did not show significant correlations. Fitness role models showed the lowest correlation. This question about the influencing factor shows a subjective picture and the related correlation. However, it is not possible to demonstrate an actual statement about the consumption of new media in connection with satisfaction with one’s own buttocks. The correlation between influencing factors and happiness with one’s own buttocks is shown in Figure 4A.

Therefore, the questioned media consumption was compared with the satisfaction with one’s own buttocks esthetics. An increasing frequency of use of Instagram (*p* < 0.001, ρ = 0.134) and TikTok (*p* = 0.009, ρ = 0.091) showed a negative correlation with the perception of one’s own buttocks, while pornography did not (*p* = 0.023, ρ = −0.079). A subjective connection between one’s own perception and new media could not be shown, but a highly significant negative effect of Instagram and TikTok on the perception of one’s own buttocks was found. This relationship can be represented as a function of frequency, i.e., the higher the consumption, the worse the self-perception of their buttocks. This correlation is shown within a matrix plot in Figure 4B.

Hypothesis 2a: The role of imagery media on the dissatisfaction with one’s own buttocks was elaborated. The study showed that subjectively naming the influencing media inputs did not necessarily correlate with their real importance. Media consumption, i.e., Instagram or TikTok, deteriorated satisfaction with one’s own buttocks.

### 3.5. Correlation of the Influencers’ WHR with the Participants’ Favored WHR

The use of social media was identified as a crucial factor for the transportation of buttocks esthetics. While there was no significance for the subjective influence of social media, the study presented good evidence for a negative correlation between media consumption as stated and happiness with one’s own buttocks. As a next step, the most followed Instagram fitness influencer within each of the target countries was identified. To be able to map the target countries of the influencers, only those with a majority of followers from their own country were included. The mean WHR was matched with the participants’ favored WHR derived from selected WHR 3D models. Influencers from Germany (0.650 vs. 0.656, *p* = 0.651), Nigeria (0.641 vs. 0.641, *p* = 0.984) and the USA (0.623 vs. 0.636, *p* = 0.147) reflected the most favored WHR chosen by the participants. In Japan, a large divergence was found between the favored WHR by participants (0.738) and the WHR of the Japanese fitness influencer (0.685, *p* = 0.004). When comparing the WHR of fitness influencers between the countries, a significant difference was found (*p* = 0.01). The favored WHR of the participants of the survey also showed a difference between the countries (*p* = 0.001). The generated 3D WHR model pictures are shown in Figure 5, and the correlation between the participants’ favored WHR and the actual Instagram influencer’s WHR is demonstrated in Table 3.

Hypothesis 2b: In addition, we assumed a stringent specification of beauty ideals of the buttocks through social media. This could be verified by matching the fitness influencers to the study participants’ preferred WHR.

## 4. Discussion

This study was able to show that women from different cultural backgrounds dealt with imperfections in their own buttocks in significantly different ways. Women in Japan showed a lower tendency to have esthetic corrections carried out, despite a clearly negative assessment of their own buttocks. A contrast between Nigeria and Germany or the USA was also shown. A positive self-assessment was noticeable in all three countries, but this nevertheless led to significantly higher requests for optimizing esthetic interventions in Germany and the USA. One reason for the discrepancy between dissatisfaction with one’s own buttocks and the urge for optimization between the countries could have been the different values stated when rating the buttocks as an attractiveness feature. The buttocks were the least relevant for Japanese women, while women in the USA considered them disproportionately important. Our study was consistent with the findings that people from countries reacted differently upon dissatisfaction with their own body [42]. One explanation for this coincides with previous studies that various beauty ideals and focuses are set in different cultures [40].

Another finding of the study was that the most important influencing factors on the satisfaction with women’s own buttocks could be identified. Satisfaction with their sex life, country of origin, weight and social media were found to be decisive for satisfaction among women. This finding supported previous studies that indicated a positive correlation between body image and sexual satisfaction [48,49,50].

It was shown that the absence of role models had a positive influence on one’s own body image of the buttocks, while naming celebrities as an image comparison had a significantly negative influence. This was consistent with previous studies [33,51]. An expected trend could be reproduced in a highly significant manner. With increasing consumption of Instagram and TikTok, dissatisfaction with women’s own buttocks increased. The negative influence of both TikTok and Instagram on the body was shown before with the limitation of asking about the whole body, not a specific anatomical feature [52,53,54].

Influencers as the main vehicles of body imagery had a great impact on the viewers’ own body image. It does not matter whether they are popular or not [55]. We found that the most followed fitness influencers across all countries included in the study had a WHR between 0.636 and 0.685 presented on their images. This corresponds to the previously mentioned unrealistic slim-thick body image [13]. This WHR of around 0.65 from influencers on Instagram is less than the previously idealized 0.7 that is mentioned in the literature [21,24,25,26]. The study participants’ preferred a WHR that broadly matched those of the fitness influencers, but not when Japan is viewed in isolation, where a participant-preferred WHR of 0.738 was found compared with the influencer’s WHR of 0.685. This was probably where the ideal-thin preference in Japan came into play.

### 4.1. Limitations and Strengths of the Study

A strength of the study was its ability to evaluate an anatomical feature, in this case, the buttocks, directly in the target group. Other studies that evaluated ideal buttocks were performed by only asking, e.g., plastic surgeons, but not the actual women [56]. A sufficiently large sample could be addressed in the target groups, although the different median ages of the countries did not entirely correspond to reality. This may have been because a web-based survey reaches a younger audience. However, according to the literature, we do not assume any substantial bias from this type of survey [57,58]. It was also reported that platforms like SurveyMonkey reach more educated and Caucasian target groups [59].

Moreover, the evaluation of the influencing variables using a machine learning approach opens up possibilities to identify previously undiscovered relationships. Especially for larger sample sizes, this results in a statistically underpinned statement like in this study. For this reason, a much more solid statement can be made based on the machine learning approach. This represents a novelty in this thematic context.

Further studies should be based on variables not considered here, such as religiosity, spirituality, self-perception in different sexualities or the influence of mental disorders on the body image of the buttocks. A bias can arise if, for example, spirituality is not taken into account.

We see another limitation in the picture series for examining the WHR, where a Caucasian female was used as the 3D model. It was reported that Black women experience a decrease in body satisfaction when exposed to media images of thin-ideal Caucasian images [60]. In future research, this should be considered, and a series of multiethnic pictures would overcome this obstacle. Nevertheless, using 3D models represents an anatomically much more sufficient methodology than line drawings or photoshopped images of people.

### 4.2. Future Directions

With the help of machine learning techniques, which have recently increased in applicability and therefore significance, it is possible to carry out better statistical processing of study results. This applies in particular to the description of relationships that are chosen arbitrarily in other methods. CARTs, random forest trees and other classification systems within machine learning capture complex relationships that should and will play an important role in attractiveness research, especially with large numbers of cases. A core statement of this study was the new concerning development of the supernormal ideal. This means that it can no longer be achieved naturally. However, this ideal is propagated with great pressure. Research should find ways to relieve this pressure, especially from young peers following this ideal. Some strategies for this have already been described and should be evaluated for their effectiveness.

## 5. Conclusions

This study has implications for our understanding of body image. The target countries showed completely different outcomes in terms of how they see themselves, how they rate their buttocks and how they cope with dissatisfaction. Furthermore, different driving forces were identified as influencing happiness with one’s own buttocks among the four regions of interest.

The data suggested social media as major forces for unhappiness with one’s own buttocks. Increased frequency of access directly correlated with dissatisfaction in women with one’s own buttocks. The study underlined the negative influence of digital media on body self-image and the translation of this harm into action, as shown with the consideration of esthetic surgery. According to this study, in addition to exposition to images on social media, celebrities also contributed to serving as role models and generating a corresponding positive influence. With the slim-thick type, a dangerous trend is now appearing. The analysis of female fitness influencers on Instagram shows the representation of a 0.7 WHR, which was consistent in all four regions. This body shape is difficult to achieve naturally. In the case of buttock operations, which have a high risk profile relative to other esthetic operations, there is a need for action, especially when plastic surgeons globally favor a similar body image [56]. Future starting points for studies can be how such trends affect the increase in esthetic interventions and exaggeratedly show what morbidity and mortality through esthetic interventions ultimately go hand in hand with the consumption of social media. In addition to further scientific studies, this study can also provide explanation potential. Actions such as promoting body positivity, strengthening self-image, movements by social media corporations in a body image-friendly direction and, finally, changed behavior of plastic surgeries should stop this current development of growing dissatisfaction with one’s own body. 

## Figures and Tables

**Figure 1 ijerph-19-13212-f001:**
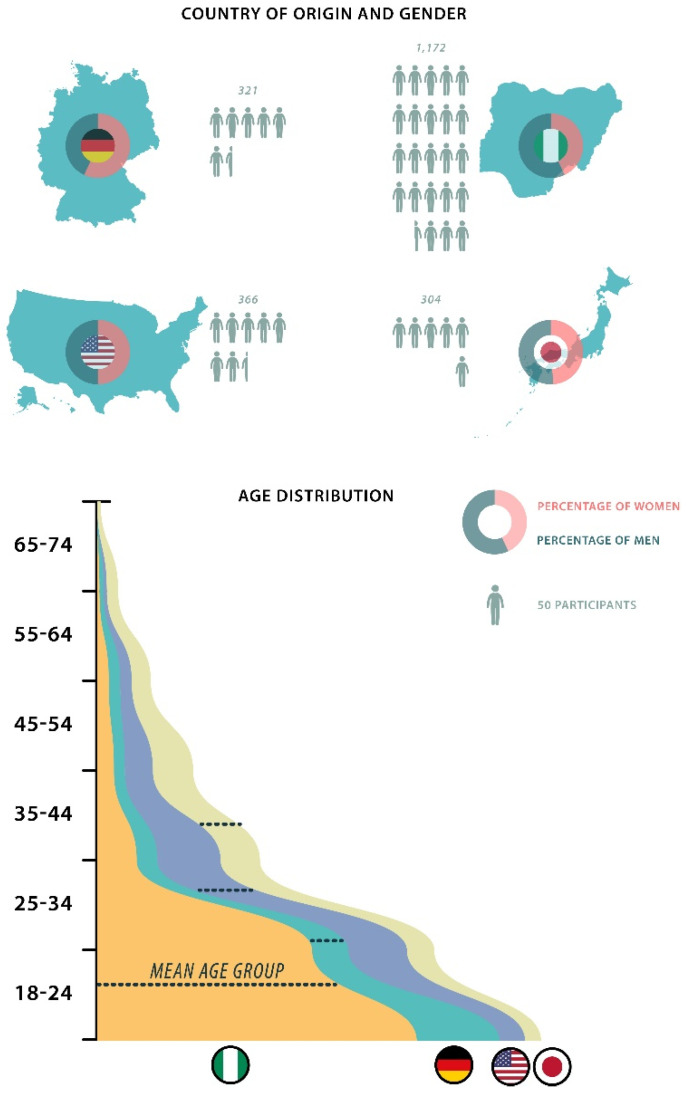
Demographic data of the complete survey population. The number of participants, their relative gender distribution and the mean age groups are shown.

**Figure 2 ijerph-19-13212-f002:**
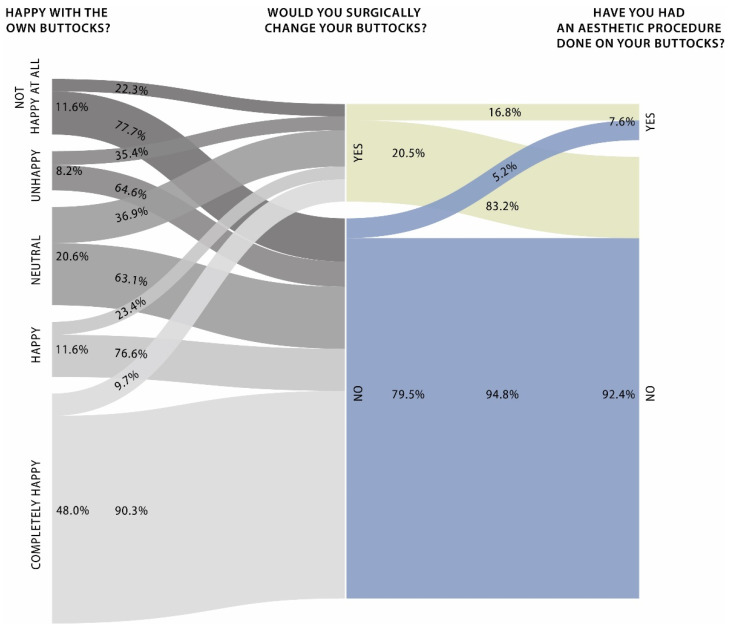
Conversion rate of happiness with one’s own buttocks to considering esthetic surgery and performing esthetic surgery. The alluvial flow diagram shows the conversion rate from the female participants’ happiness about their own buttocks to considering esthetic procedures on the buttocks to the performed esthetic procedures on the buttocks. *n* = 963 female participants.

**Figure 3 ijerph-19-13212-f003:**
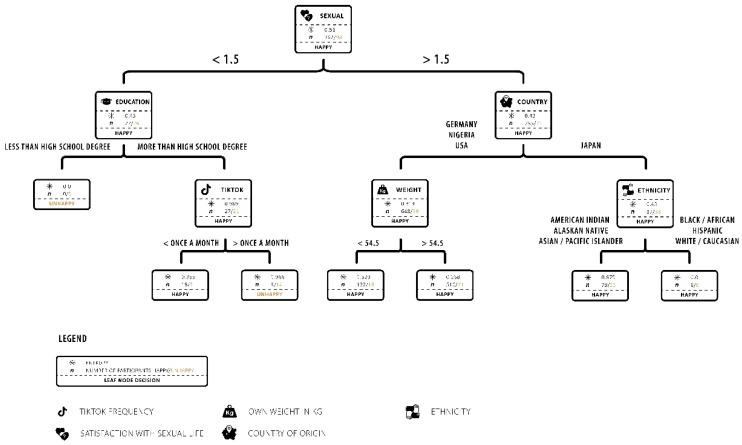
Factors influencing happiness with one’s own buttocks in women. A classification and regression tree (CART) describes the most important factors that influence the evaluation of happiness with one’s own buttocks (binary value: happy and unhappy). The CART depicts the splitting variables, the entropy, happy, unhappy and the number of survey participants. *n* = 963 female participants.

**Figure 4 ijerph-19-13212-f004:**
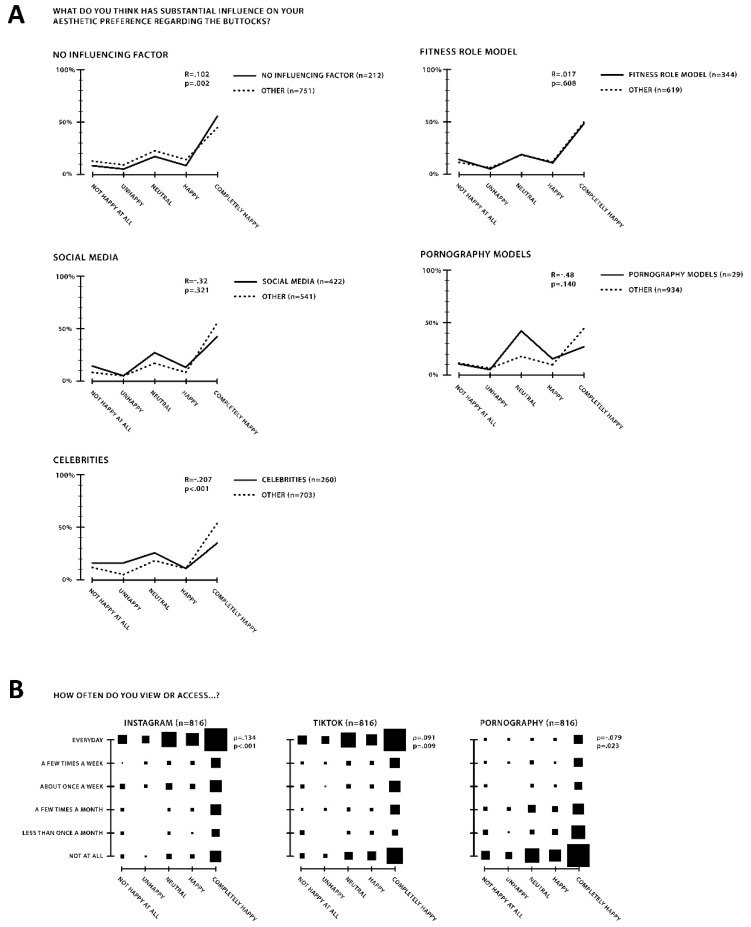
Influence of social media on happiness with one’s own buttocks in women. (**A**) Correlation of happiness with one’s own buttocks and the role models that influence the gluteal esthetics of the participants. Female participants were asked who their most influential role models were regarding gluteal esthetics. Multiple responses were possible: 212 (22%) participants stated no specific influencing factor, 422 (44%) social media, 260 (27%) celebrities, 344 (36%) fitness role models and 29 (<1%) pornography. A correlation was found between happiness and specific influencing factors with no specific role model (R = 0.102, *p* = *0*.002, ρ = 0.106) and celebrities (R = −0.207, *p* < 0.001, ρ = −0.218). Graphs show the percentage within each group and their distribution with respect to happiness. (**B**) Sizes of the matrix plot depict the distribution of happiness correlating with social media consumption. Participants were asked how often they accessed Instagram, TikTok and pornography. The frequency was correlated with the stated happiness with one’s own buttocks. Significant correlations were found between happiness and consumption frequency for all three media types: Instagram (*p* < 0.001, ρ = 0.134), TikTok (*p* = 0.009, ρ = 0.091) and pornography (*p* = *0*.023, ρ = −0.079). The *p*-value is stated as the level of significance calculated using the Pearson correlation factor (R) and Spearman rank correlation (ρ). *n* = 963 female participants.

**Figure 5 ijerph-19-13212-f005:**
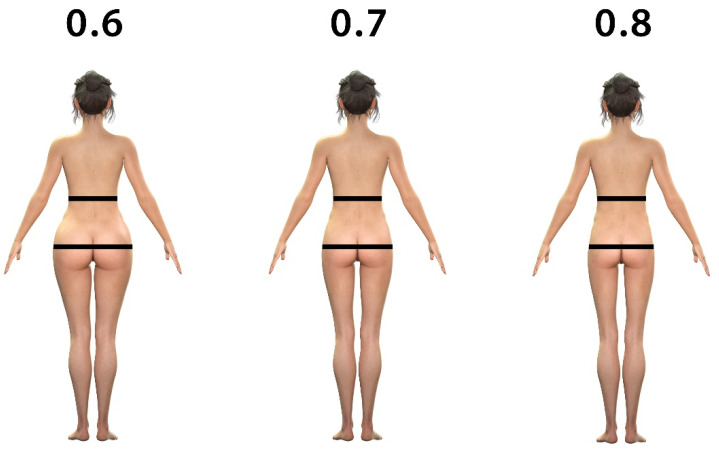
Generated WHR models for the survey. The black bars on the 3D images show the measurement points for the WHR.

**Table 1 ijerph-19-13212-t001:** Perception of one’s own buttocks in the female participants.

	Germany (*n* = 151)		Nigeria (*n* = 486)		USA (*n* = 179)		Japan (*n* = 144)		*p*-Value
	* **M** *	* **SD** *	* **M** *	* **SD** *	* **M** *	* **SD** *	* **M** *	* **SD** *	
Happy with one’s own buttocks?	3.45	1.13	4.37	1.25	3.66	1.22	2.18	1.00	0.001
Would you surgically change your buttocks? (%)	23.8		8.2		63.1		4.3		0.001
Have you had an esthetic procedure done on your buttocks? (%)	17.2		5.3		11.2		1.3		0.001
What part of the female body are you most attracted to? (%)									
Breast	21.2		20.0		28.5		27.9		0.001
Buttocks	19.9		34.4		57.0		5.4		0.001
Abdomen	10.6		5.3		0.0		2.7		0.001
Legs	8.6		1.9		0.0		23.1		0.001
Face	39.7		38.6		14.5		40.8		0.001
Total	100		100		100		100		

Note. *M* and *SD* represent the mean and standard deviation, respectively. The *p*-value is stated as the level of significance found after performing an ANOVA followed by a multiple test comparison via Tukey’s post hoc test (homoskedasticity) or a Brown–Forsythe and Welch ANOVA followed by Dunnett’s T3 post hoc test (heteroskedasticity) for multi-group analysis.

**Table 2 ijerph-19-13212-t002:** CART analysis of factors that influenced happiness with one’s own buttocks.

Feature	Variable Feature Importance Score (a.u.)
Satisfaction with sex life	0.405
Country of origin	0.353
Weight	0.096
Level of education	0.061
TikTok frequency	0.042
Ethnicity	0.042

Note. Training accuracy was 0.898 and testing accuracy was 0.875.

**Table 3 ijerph-19-13212-t003:** Comparison between Instagram fitness influencer WHRs and the participants’ favored WHRs.

	Germany	Nigeria	USA	Japan	*p*-Value
Participants’ favored WHR (a.u.)	0.650 (*n* = 275)	0.641 (*n* = 1057)	0.623 (*n* = 375)	0.738 (*n* = 304)	0.001
WHR of fitness influencer (a.u.)	0.656 (*n* = 25)	0.641 (*n* = 11)	0.636 (*n* = 49)	0.685 (*n* = 17)	0.001
*p*-value	0.651	0.984	0.147	0.004	

Note: ANOVA followed by multiple test comparison via Tukey’s post hoc test (homoskedasticity) or Brown–Forsythe and Welch ANOVA followed by Dunnett’s T3 post hoc test (heteroskedasticity) for multi-group analysis. For a comparison of two groups, the significance was calculated with Student’s *t*-test. *n* = 2011 participants.

## Data Availability

The study data is available upon request.

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
