# Peer review of "Interethnic Influencing Factors Regarding Buttocks Body Image in Women from Nigeria, Germany, USA and Japan"

_ijerph, 2022, doi:10.3390/ijerph192013212_

Round 1

Reviewer 1 Report

In this manuscript, the authors have investigated differences between nationalities in the perception of buttocks in women.

However, the article has several serious weaknesses, which I list below. My intention as a reviewer is none other than to improve the work you have submitted for publication. Your work is powerful, but it is not at all well-targeted.

Abstract. No numerical results are given for all statistical tests performed. Include conclusions of results in the results section, this is only in the discussion of the article.

Advice: Without being strict, but as an idea of text distribution.

350 words.

Introduction (including objective) 100 words

Methods 50 words

Results 150 words, with data!

Conclusion: 50 words

Introduction The introduction is too, too long, longer that any journal allowed. In this section, aspects are mentioned and discussed far from the objective of the study and digresses into different social media trends. The four hypotheses should be summarized in a single, simpler hypothesis. The hypotheses stated afterward are not the same as the results. For example, hypothesis 2 in the introduction talks about psychological aspects that are not supported by the results.

Advice: Focus on the objective. Introduction structure.

What is the problem/aspects that you could solve with your results. The magnitude of the problem and why it is relevant.

What is the knowledge of this problem at this moment, what are the aspects not known, and what is your objective, which will contribute to improving current knowledge.

Methods

A web-based questionnaire is not a study design, it is the tool in which you collect the data. The study design is a cross-sectional study, because there is no follow-up of the patient. In statistical methods, only describe the software use for it, but there is no description of the statistical methods (different test used and the bases of their applications) In the results section or in the figure legend, there appears Spearman correlation; the ANOVA test and post hoc tests, all of this must be presented at the methods section. Also, must be described transformation of variable performed, like how the dimensionality of the variable happiness was reduced from 5 stars scale to yes and no (1, 0), it is enunciated but not described (a 0 is less than 2, 3 or 4?).

Results

Survey Population

- Exclude everything other than women

- Order results from maximum to minimum or minimum to maximum, but order is better for readers

- Possible uncontrolled bias, age with use of social media Tick-Tock. Tick-Tock is a social media with the youngest audience.

Perception of the own buttocks in women

 Figure 1 does not contribute anything to the results; consider eliminating it.

- It is described how quantitative variables are evaluated (ANOVA), but not qualitative variables.

Identification of major Factors Influencing the Perception of the own Buttocks

 The variables included in the tree are not indicated how they are coded. This should be described in the methods.

- Possible bias. It is not described whether the religion they process and whether they are practicing is included among the participants, as this aspect can greatly modify the results.

Social media is identified as a Crucial Trigger for Perception of Gluteal Aesthetics in Females

 The statistical method used (Pearson and Spearman correlation) is not suitable for the analysis of these results. Moreover, only one of the methods is valid if the application conditions are evaluated. In addition to the fact that the method used is inconsistent, the magnitude of the correlation coefficients are so low that no conclusions can be drawn.

 Uncontrolled bias. Social media usage in general, without specifying the content consumed, is a very biased measure to indicate that it is a "crucial trigger". I may be a heavy consumer of cat videos, which I believe do not influence the perception of my buttok.

Influencers WHR in coherence with participants favored WHR

                Eliminate non-female respondents because this is a major bias in the WHR measures. Table 3 presents the total number of respondents.

Conclusions

The conclusions drawn in the article do not correspond to the results obtained.

Tables.

In all table titles the first letter are missing. These details give a bad impression to any reviewer (lack of review by the authors). Inappropriate figure legend

Figures

The figures are barely visible, specially figures 2 and 3. It would be advisable for the authors to improve the resolution of these images. Only figure 4 has an acceptable quality.

Author Response

See letter attached

Reviewer 2 Report

According to the authors, the objective of this study was to broaden the literature about the perception of the female buttocks with the implication of social media influence. In a general way, the study was well conducted, mainly by the strategy used (web-based questionnaire) and also by the statistical analysis, and it presented interesting results. However, there is a couple of points that should be concerned.

In the Abstract section:

1) Please, provide the average age and standard deviation of the all sample.

2) Please, add the statistical significance found in the study for relevant results described in this section.

3) Line 30 “high level of dissatisfaction with their own.Conclusion: Overall, a highly significant correlation was”, please correct the spacing errors.

In the Introduction section:

4) Considering that in the study a lot of emphasis is placed on Media and images, it would be helpful to include a brief description of image-based Social Media, such as Instagram.

5) On page 2, lines 53, the authors cited that "With the Thinspiration trend on social media a certain body picture was enforced.”. Please, cite references.

6) On page 2, lines 97 to 99, the authors cited that “Celebrities achieve these forms only through aesthetic procedures such as breast augmentation, lip-osuction of the waist, but especially buttock augmentation.” Please, could you mention any Celebrities who have formalized these aesthetic procedures?

7) On pages 4, in the Hypotheses section, it is suitable to assume a link between the first two hypotheses and the last two by renaming them H1a, H1b, H2a and H2b.

In the Material and Methods section:

8) Please, add the number of Ethical approvals of the study in the Study Design section.

9) On pages 4, in the Study Design section, the authors cited that “The first part contained general questions about 168 demography, level of education, relationship status, social media consumption as well as 169 the feeling about the own buttocks, own body preference as well as the attitude towards 170 aesthetic interventions”, it is suitable to assume a table with these demographic characteristics of the sample.

10) In the Study Design section, the authors do not mention the use of an Informed Consent by participants. Has it been included in the study?

In the Results section

11) On page 5, lines 206 to 207, the authors cited “Participants from Germany (n = 321), Nigeria (n = 1,172), USA (n = 366) and Japan (n 206 = 304)”, it is suitable to add the average of the age and SD for each sample.

12) On page 8, lines 301 to 303, the authors cited “212 female participants expressed no specific influence on their personal gluteal perception. 422 female participants identified social media, 260 celebrities, 344 fitness role models and 29 models from pornography as their main influence for the own gluteal perceptionit is suitable to add the percentage instead of the frequency.

In the Discussion section:

13) Although the discussion highlights the main findings of the present study, it is suitable to broaden the interpretation of these results, also referring to previous literature and emphasizing the importance and innovativeness of this study

14) Some Strengths and Limitations should be added.

15) On pages 11, in the Limitations, Strengths and Future Directions section, please insert and expand the Future Directions of research on this topic which were instead included in the conclusions.

Author Response

See letter attached

Reviewer 3 Report

The subject under study is of great relevance although it focuses on a single part of the body. Despite this, the methodology used, the results and the conclusions are very well formulated and respond to the research objectives and hypotheses.

To improve the understanding of some parts and improve the quality of the article, please add some comments:

1. Add in the summary the % of total men and women, as well as the average age.

2. In the methodology section, the questions asked would be explained in more detail and a final annex would be added with the questionnaire that was passed exactly. It would also be good, if possible, to do the Cronbach's alpha of the items used to measure each subcategory.

3. Reorder the conclusions according to the order of the hypotheses raised.

Good work.

Author Response

See letter attached

Round 2

Author Response

See the file attached
